# Association of red cell distribution width and its changes with the 30-day mortality in patients with acute respiratory failure: An analysis of MIMIC-IV database

Yuyi Shi[1], Liuxian Shi[1], Fei Chen[1], Zhipeng Jiang[2], Kaihui Sheng[3]*

1 Department of Emergency, Shanghai Minhang District Medical Emergency Center, Shanghai, P.R. China,
2 Department of Administrative Management, Shanghai Minhang District Medical Emergency Center,
Shanghai, P.R. China, 3 Department of Emergency Response and Medical Training, Shanghai Minhang
District Medical Emergency Center, Shanghai, P.R. China

* shengkaihuish@outlook.com

## Abstract

### Background

Acute respiratory failure (ARF) is a common disease in the intensive care units (ICUs) with high risk of mortality. The red cell distribution width (RDW) is one of baseline ICU indicators which can be easily available, and has been used in the long-term prognostic analyses of diseases. However, no studies have explored the role of baseline RDW and its change during hospitalization in in-hospital mortality in ARF. Herein, this study aims to explore the association between RDW and its changes and the 30-day mortality in ARF patients.

### Methods

Demographic and clinical data of 7,497 patients with ARF were extracted from the Medical Information Mart for Intensive Care IV (MIMIC-IV) database in 2012–2019 in this retrospective cohort study. Univariable and multivariable Cox regression analyses were used to explore the association between RDW and its changes and 30-day mortality with hazard ratios (HRs) and 95% confidence intervals (CIs). Subgroup analyses of different baseline RDW levels were also performed. We then assessed the predictive performance of RDW changes combined with the Sequential Organ Failure Assessment (SOFA) score on 30-day mortality using receiver operator characteristic curves (ROCs) with areas under curve (AUCs).

### Results

Totally, 2,254 (30.07%) patients died in 30 days. After adjusting for covariates, we found that high baseline RDW [HR = 1.25, 95%CI: (1.15–1.37)] and RDW changes ≥0.3% [HR = 1.12, 95%CI: (1.01–1.24)] were both related to an increased risk of 30-day mortality. In patients whose baseline RDW level ≥14.9%, RDW changes ≥0.3% was also associated with an increased risk of 30-day mortality [HR = 1.19, 95%CI: (1.05–1.35)]. Moreover, the

**Data Availability Statement:** The datasets generated and/or analyzed during the current study

are available in the MIMIC-IV database, https://mimic.physionet.org/iv/.

**Funding:** The author(s) received no specific funding for this work.

**Competing interests:** The authors have declared that no competing interests exist.

predictive value of RDW changes combined with SOFA on 30-day mortality was a little better than that of single SOFA score, with AUCs of 0.624 vs. 0.620.

## Conclusion

High baseline RDW level and its changes during hospitalization was relate to the increased risk of 30-day mortality in ARF, and the predictive value of RDW changes for ARF short-term mortality is still needed exploration.

## Introduction

Acute respiratory failure (ARF) is a common clinical critical syndrome along with damage to the function of multiple organs, such as the heart, kidney, and brain [1]. ARF is one of the common diseases in intensive care units (ICUs), and about 36% of patients with ARF die during hospitalization [2]. Due to the high in-hospital mortality, finding predictors related to the risk of short-term mortality in ARF is essential for its prognostic management.

The red cell distribution width (RDW) reflects systemic inflammatory state and oxidative stress, and is easily available in the standard complete blood cell count [3]. RDW values have been used in the prognostic analyses of many severe diseases [4]. A recent study by Zhang et al. [5] showed that high level of RDW was related to an increased risk of long-term mortality in patients with ARF during the 3 years follow-up. Comparing to the long-term mortality, which are influenced by multiple out-of-hospital factors, baseline ICU indicators including RDW are used more often in the prediction of short-term mortality outcomes [6, 7]. Moreover, it is of concern that the blood indicators of critically ill patients are in rapid and dynamic change due to factors such as changes in their condition and treatment, which may affect their prognosis [8, 9]. Foy et al. [10] investigated the patients with COVID-19, and found that elevated RDW at the time of hospital admission and an increase in RDW during hospitalization were both associated with increased mortality risk. Another study in patients with community-acquired pneumonia (CAP) have found an increased risk of short- and long-term mortality in patients with greater elevations or decreases in RDW [11]. However, no studies have reported the relationship between baseline RDW and its change in value during hospitalization and the risk of mortality in ARF.

Herein, this study aims to explore the association between RDW at the admission and its changes during hospitalization of ICU and the risk of 30-day mortality in patients with ARF. We also assessed the predictive performance of RDW changes on the 30-day mortality, and hope to provide some references for finding a convenient tool to identify patients with ARF in poor prognoses.

## Methods

### Study design and participants

Data of participants in this retrospective cohort study were extracted from the Medical Information Mart for Intensive Care IV (MIMIC-IV) database. The MIMIC database is jointly published by the computational physiology laboratory of Massachu-setts Institute of Technology (MIT, Cambridge, MA, USA), Beth Israel Deaconess Medical Center (BIDMC, Boston, MA, USA), and Philips Medical. The clinical diagnosis and treatment information on more than 40,000 real patients who are predominantly White people living in the ICU of the BIDMC

were collected and sorted out by MIMIC database since 2001. More details of the public data are available on the website: https://mimic.mit.edu/.

A total of 10,945 adults were diagnosed with ARF in MIMIC-IV database in 2012–2019. The exclusion criteria were (1) aged <18 years old, (2) missing information of RDW at 24 hours after the admission of ICU (T0) and the 24 to 48 hours during ICU hospitalization (T1), and (3) hospitalized in the ICU for less than 48 hours. After excluding those who have not meet the inclusion criteria, 7,497 of them were eligible. The MIMIC was approved by the Institutional Review Boards (IRBs) of BIDMC and the MIT. Since the MIMIC-IV database was publicly available, written informed consent from participants was obtained before data collection, and all the data of participants were de-identified, no ethical approval was needed by the IRB of Shanghai Minhang District Medical Emergency Center. In addition, this study was performed in accordance with the Strengthening the Reporting of Observational studies in Epidemiology (STROBE) reporting guideline.

## Diagnosis of acute respiratory failure

ARF diagnosis was according to the international classification of diseases-9 (ICD-9) code "51881." For patients readmitted to the same hospital, only information from the first hospitalization with ARF diagnosis was retained [12].

## Variable collection

This study was conducted with the MIMIC-IV database using Structured Query Language (SQL) and PostgreSQL software (version 9.6.22). Variables were all collected within the first 24 hours after patients entering the ICU, including age, gender, ethnicity, type of ICU at first hospitalization, ventilation time, vasopressors use, renal replacement therapy (RRT), antiplatelet use, antibiotics use, blood transfusions, atrial fibrillation, pneumonia, acute kidney injury (AKI), sepsis, $SpO_2$, systolic blood pressure (SBP), diastolic blood pressure (DBP), mean arterial pressure (MAP), temperature, heart rate, respiratory rate (RR), weight, $FIO_2$, hemoglobin (HB), blood urea nitrogen (BUN), platelet, white blood cell (WBC), hematocrit, lactate, glucose, creatinine (Cr), bicarbonate, international normalized ratio (INR), sodium (Na), potassium (K), chloride, prothrombin time (PT), partial thromboplastin time (PTT), sequential organ failure assessment (SOFA) score, Charlson comorbidity index (CCI), RDW T0, RDW T1, RDW changes, and length of stay (LOS).

We extracted the RDW values which examined at two time points. "RDW T0" represented to the RDW value first examined within the 24 hours after ICU admission, and "RDW T1" represented to the RDW value first examined during the 24 to 48 hours after ICU admission. The RDW changes was calculated by RDW T1 minus RDW T0. We also classified the RDW T0, RDW T1, and RDW changes into categorical variables in the analyses. The cutoff values of RDW T0 (14.9%) and RDW T1 (15.1%) were their medians, while that of the RDW changes was the tertiles, including <0%, 0%-0.29%, and ≥0.3%.

## Outcome and follow-up

The study outcome was 30-day mortality. The MIMIC followed up by information in the electronic medical charts and hospital department records, or making contact with the patients, their family members, their attending health care workers, or family physicians on the phone. The follow-up started after the 24 hours of the first ICU admission at the first time after hospitalization, and ended when patients died or 30 days after the admission of ICU.

## Statistical analysis

Normal distributed data were described by mean ± standard deviation (mean ± SD), and using t test for comparison between groups. Skewed distribution data were described by median and quartiles [M (Q1, Q3)], and using Wilcoxon rank sum test for comparison. The frequency and composition ratio [N (%)] was used to describe the distribution of measurement data, and chi-square test was used for comparison.

Univariable Cox regression and bidirectional stepwise regression analyses were used to screen the covariates. Covariates that significantly associated with the 30-day mortality ($P<0.05$) were included in the adjustment of multivariable model. The relationship between RDW (T0 and T1), RDW changes and the 30-day mortality in patients with ARF was explored using univariable and multivariable Cox regression analyses. The evaluation indexes were hazard ratios (HRs) and 95% confidence intervals (CIs). Then we explored these associations in different RDW T0 level subgroup. We also drew receiver operator characteristic curves (ROCs) with areas under curve (AUCs) to assess the predictive performance of RDW changes combined with SOFA on the 30-day mortality in ARF. The SOFA score is a validated prognostic score ranging from 0–24, with points assigned for evidence of organ failure within 6 different organ systems, with higher scores correlating with a higher likelihood of in-hospital mortality [13].

Two-sided $P<0.05$ was considered significant. Missing variables were deleted if their proportion over 20%, otherwise were interpolated by the random forest method [14]. Sensitivity analysis of the participants' characteristics before and after the interpolation was showed in S1 Table. Statistical analyses were by Python 3.9.12 (Python Software Foundation, Delaware, USA) and SAS 9.4 (SAS Institute., Cary, NC, USA).

## Results

### Characteristics of participants

Fig 1 was the flowchart of participants screening. A total of 10,945 adult patients with ARF were initially included. We excluded those who without the information of RDW (T0 or T1) (n = 1349), or hospitalized in the ICU for less than 48 hours (n = 2099). Finally, 7,497 of them were eligible. S1 Table showed the sensitivity analysis of the participants' characteristics before and after the interpolation of missing variables, and no significant difference was found.

Table 1 showed the characteristics of eligible ARF patients. Among them, 2,254 (30.07%) died within 30 days. The average age of the participants was 64.09 years old, and 4,188 (55.86%) of them were males. We found that between survival group and 30-day mortality group, age, ethnicity, vasopressors use, RRT, antibiotics use, blood transfusions, atrial fibrillation, AKI, sepsis, SBP, DBP, MAP, temperature, RR, weight, $FIO_2$, HB, BUN, platelet, WBC, hematocrit, lactate, Cr, bicarbonate, INR, chloride, PT, PTT, SOFA, CCI, RDW T0, RDW T1, RDW changes, LOS, and follow-up time were all significantly different, indicating they were potential covariates associated with the in-hospital mortality (all $P<0.05$).

### Relationship between RDW and its changes and 30-day mortality in acute respiratory failure

We first screened the covariates associated with the 30-day mortality in patients with ARF (Table 2). The results showed that age, ethnicity, type of ICU at first hospitalization, ventilation time, vasopressor use, RRT, antibiotics use, AKI, temperature, RR, weight, $FIO_2$, BUN, lactate, Cr, INR, chloride, PTT, and CCI were significantly associated with 30-day mortality, and they were further included in the adjustment for multivariable models (all $P<0.05$).

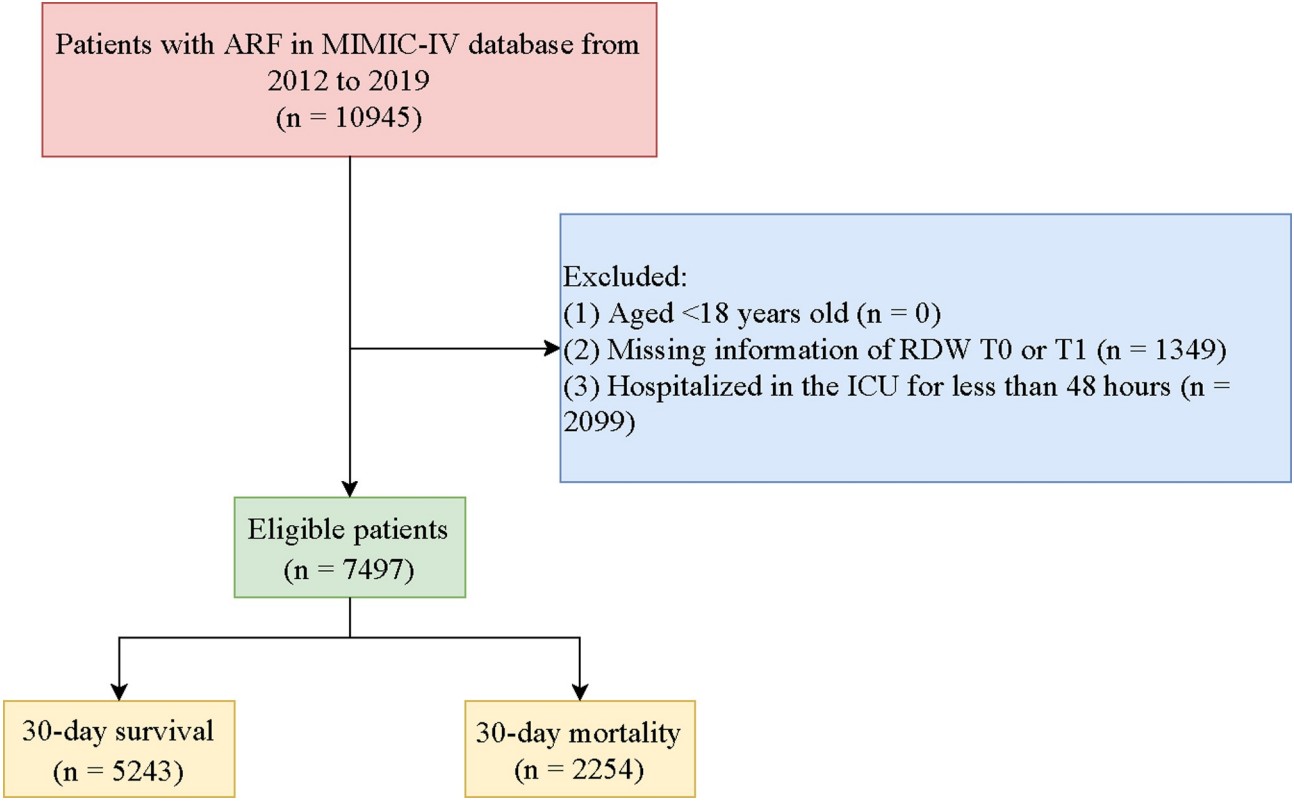

**Fig 1. Flow chart of participants screening.**

The associations between RDW T0 and 30-day mortality, and between RDW changes and 30-day mortality were showed in Table 3. After adjusting for covariates, we found that RDW T0 [HR = 1.07, 95%CI: (1.05–1.09)] and RDW changes [HR = 1.08, 95%CI: (1.02–1.13)] were both associated with the 30-day mortality in patients with ARF. Compared with RDW T0 <14.9%, RDW T0 ≥14.9% was associated with higher risk of 30-day mortality [HR = 1.25, 95%CI: (1.15–1.37)]. Similarly, patients with RDW changes ≥0.3% seemed to have higher risk of 30-day mortality compared with those who with RDW changes <0% [HR = 1.12, 95%CI: (1.01–1.24)].

## Association between RDW changes and 30-day mortality in subgroups of RDW T0 levels

Table 4 showed the results of subgroup analysis. In RDW T0 <14.9% subgroup, we have not found any significant association between RDW changes and the risk of 30-day mortality (all $P>0.05$). However, in RDW T0 ≥14.9% subgroup, compared with RDW changes <0%, RDW changes ≥0.3% was associated with higher risk of 30-day mortality [HR = 1.19, 95% CI: (1.05–1.35)].

## The predictive performance of RDW changes on 30-day mortality in different RDW T0 levels

We further explored the predictive performance of RDW changes on 30-day mortality in different RDW T0 level subgroups (Tables 5 and 6). Fig 2 showed the ROC curves of the

**Table 1. Characteristics of ARF patients.**

| Variables | Total (n = 7497) | Survival (n = 5243) | 30-day mortality (n = 2254) | Statistics | P |
|---|---|---|---|---|---|
| Age, years, Mean ± SD | 64.09 ± 16.28 | 62.03 ± 16.46 | 68.88 ± 14.78 | t = -17.77 | <0.001 |
| Gender, n (%) | | | | $\chi^2 = 0.304$ | 0.582 |
| Female | 3309 (44.14) | 2325 (44.34) | 984 (43.66) | | |
| Male | 4188 (55.86) | 2918 (55.66) | 1270 (56.34) | | |
| Ethnicity, n (%) | | | | $\chi^2 = 36.000$ | <0.001 |
| White | 4758 (63.47) | 3353 (63.95) | 1405 (62.33) | | |
| Black | 749 (9.99) | 566 (10.80) | 183 (8.12) | | |
| Others | 917 (12.23) | 647 (12.34) | 270 (11.98) | | |
| Unknown | 1073 (14.31) | 677 (12.91) | 396 (17.57) | | |
| Type of ICU at first hospitalization, n (%) | | | | $\chi^2 = 5.225$ | 0.073 |
| MICU | 2305 (30.75) | 1575 (30.04) | 730 (32.39) | | |
| SICU | 3848 (51.33) | 2703 (51.55) | 1145 (50.80) | | |
| Others | 1344 (17.93) | 965 (18.41) | 379 (16.81) | | |
| Ventilation time, hours, M (Q$_1$, Q$_3$) | 71.75 (37.82, 136.47) | 71.80 (38.17, 139.00) | 71.72 (36.40, 132.00) | Z = -1.856 | 0.063 |
| Vasopressor use, n (%) | | | | $\chi^2 = 145.918$ | <0.001 |
| No | 3197 (42.64) | 2473 (47.17) | 724 (32.12) | | |
| Yes | 4300 (57.36) | 2770 (52.83) | 1530 (67.88) | | |
| RRT, n (%) | | | | $\chi^2 = 69.100$ | <0.001 |
| No | 6349 (84.69) | 4559 (86.95) | 1790 (79.41) | | |
| Yes | 1148 (15.31) | 684 (13.05) | 464 (20.59) | | |
| Antiplatelet use, n (%) | | | | - | 0.548 |
| No | 7484 (99.83) | 5235 (99.85) | 2249 (99.78) | | |
| Ye | 13 (0.17) | 8 (0.15) | 5 (0.22) | | |
| Antibiotics use, n (%) | | | | $\chi^2 = 22.668$ | <0.001 |
| No | 818 (10.91) | 631 (12.04) | 187 (8.30) | | |
| Yes | 6679 (89.09) | 4612 (87.96) | 2067 (91.70) | | |
| Blood transfusions, n (%) | | | | $\chi^2 = 48.476$ | <0.001 |
| No | 4391 (58.57) | 3207 (61.17) | 1184 (52.53) | | |
| Yes | 3106 (41.43) | 2036 (38.83) | 1070 (47.47) | | |
| Atrial fibrillation, n (%) | | | | $\chi^2 = 63.056$ | <0.001 |
| No | 5459 (72.82) | 3958 (75.49) | 1501 (66.59) | | |
| Yes | 2038 (27.18) | 1285 (24.51) | 753 (33.41) | | |
| Pneumonia, n (%) | | | | $\chi^2 = 2.190$ | 0.139 |
| No | 4678 (62.40) | 3300 (62.94) | 1378 (61.14) | | |
| Yes | 2819 (37.60) | 1943 (37.06) | 876 (38.86) | | |
| AKI, n (%) | | | | $\chi^2 = 102.000$ | <0.001 |
| No | 1056 (14.09) | 878 (16.75) | 178 (7.90) | | |
| Yes | 6441 (85.91) | 4365 (83.25) | 2076 (92.10) | | |
| Sepsis, n (%) | | | | $\chi^2 = 6.275$ | 0.012 |
| No | 4116 (54.90) | 2928 (55.85) | 1188 (52.71) | | |
| Yes | 3381 (45.10) | 2315 (44.15) | 1066 (47.29) | | |
| SpO$_2$, %, Mean ± SD | 96.74 ± 3.63 | 96.76 ± 3.62 | 96.71 ± 3.64 | t = 0.51 | 0.608 |
| SBP, mmHg, Mean ± SD | 123.26 ± 24.60 | 124.09 ± 24.61 | 121.33 ± 24.48 | t = 4.46 | <0.001 |
| DBP, mmHg, Mean ± SD | 68.78 ± 17.65 | 69.33 ± 17.51 | 67.49 ± 17.92 | t = 4.13 | <0.001 |
| MAP, mmHg, Mean ± SD | 86.94 ± 17.76 | 87.58 ± 17.68 | 85.44 ± 17.86 | t = 4.80 | <0.001 |
| Temperature,˚C, Mean ± SD | 36.82 ± 0.81 | 36.88 ± 0.81 | 36.66 ± 0.80 | t = 10.97 | <0.001 |
| Heart rate, bpm, Mean ± SD | 93.33 ± 21.30 | 93.09 ± 21.41 | 93.88 ± 21.04 | t = -1.47 | 0.142 |

(*Continued*)

**Table 1.** (Continued)

| Variables | Total (n = 7497) | Survival (n = 5243) | 30-day mortality (n = 2254) | Statistics | P |
|---|---|---|---|---|---|
| RR, bpm, Mean ± SD | 20.75 ± 5.97 | 20.57 ± 5.88 | 21.19 ± 6.16 | t = -4.07 | <0.001 |
| Weight, kg, Mean ± SD | 82.07 ± 20.72 | 83.30 ± 20.98 | 79.22 ± 19.80 | t = 8.03 | <0.001 |
| $FIO_2$, %, M ($Q_1$, $Q_3$) | 60.00 (50.00, 100.00) | 60.00 (50.00, 100.00) | 62.68 (50.00, 100.00) | Z = 2.896 | 0.004 |
| HB, g/dL, Mean ± SD | 10.66 ± 2.35 | 10.77 ± 2.36 | 10.39 ± 2.30 | t = 6.59 | <0.001 |
| BUN, mg/dL, M ($Q_1$, $Q_3$) | 23.00 (15.00, 40.00) | 21.00 (14.00, 36.00) | 29.00 (18.00, 47.00) | Z = 14.798 | <0.001 |
| Platelet, K/uL, M ($Q_1$, $Q_3$) | 193.00 (131.00, 262.00) | 196.00 (138.00, 261.00) | 186.50 (115.00, 265.00) | Z = -4.482 | <0.001 |
| WBC, K/uL, M ($Q_1$, $Q_3$) | 11.80 (8.20, 16.60) | 11.60 (8.00, 16.00) | 12.30 (8.60, 17.70) | Z = 5.331 | <0.001 |
| Hematocrit, %, Mean ± SD | 32.68 ± 7.02 | 32.97 ± 7.05 | 32.02 ± 6.90 | t = 5.38 | <0.001 |
| Lactate, mmol/L, M ($Q_1$, $Q_3$) | 1.79 (1.30, 2.60) | 1.70 (1.27, 2.45) | 2.00 (1.50, 3.10) | Z = 13.621 | <0.001 |
| Glucose, mg/dL, M ($Q_1$, $Q_3$) | 136.00 (108.00, 179.00) | 135.00 (108.00, 178.00) | 138.00 (108.00, 184.00) | Z = 1.331 | 0.183 |
| Cr, mg/dL, M ($Q_1$, $Q_3$) | 1.10 (0.80, 1.80) | 1.10 (0.80, 1.70) | 1.30 (0.80, 2.10) | Z = 8.527 | <0.001 |
| Bicarbonate, mEq/L, Mean ± SD | 22.33 ± 5.51 | 22.57 ± 5.49 | 21.76 ± 5.51 | t = 5.86 | <0.001 |
| INR, ratio, M ($Q_1$, $Q_3$) | 1.30 (1.20, 1.60) | 1.28 (1.10, 1.50) | 1.40 (1.20, 1.80) | Z = 11.993 | <0.001 |
| Na, mEq/L, Mean ± SD | 138.31 ± 6.07 | 138.34 ± 5.81 | 138.23 ± 6.62 | t = 0.64 | 0.519 |
| K, mEq/L, Mean ± SD | 4.25 ± 0.84 | 4.24 ± 0.84 | 4.28 ± 0.84 | t = -1.91 | 0.056 |
| Chloride, mEq/L, Mean ± SD | 103.55 ± 7.21 | 103.67 ± 6.96 | 103.26 ± 7.75 | t = 2.16 | 0.031 |
| PT, seconds, M ($Q_1$, $Q_3$) | 14.30 (12.80, 17.50) | 14.22 (12.70, 16.60) | 14.90 (13.20, 20.00) | Z = 11.945 | <0.001 |
| PTT, seconds, M ($Q_1$, $Q_3$) | 32.40 (27.80, 40.60) | 31.80 (27.60, 39.35) | 34.21 (28.40, 43.90) | Z = 8.395 | <0.001 |
| SOFA, score, M ($Q_1$, $Q_3$) | 8.00 (5.00, 11.00) | 7.00 (5.00, 10.00) | 9.00 (6.00, 12.00) | Z = 16.121 | <0.001 |
| CCI, score, M ($Q_1$, $Q_3$) | 3.00 (2.00, 5.00) | 3.00 (1.00, 5.00) | 4.00 (2.00, 6.00) | Z = 16.706 | <0.001 |
| RDW T0, %, Mean ± SD | 15.52 ± 2.48 | 15.22 ± 2.28 | 16.20 ± 2.78 | t = -14.67 | <0.001 |
| RDW T1, %, Mean ± SD | 15.73 ± 2.47 | 15.41 ± 2.25 | 16.46 ± 2.80 | t = -15.75 | <0.001 |
| RDW changes, %, M ($Q_1$, $Q_3$) | 0.20 (-0.10, 0.40) | 0.10 (-0.10, 0.40) | 0.20 (-0.10, 0.50) | Z = 3.850 | <0.001 |
| LOS, days, M ($Q_1$, $Q_3$) | 5.99 (3.54, 11.13) | 5.99 (3.48, 11.70) | 5.92 (3.66, 10.13) | Z = -2.139 | 0.032 |
| Follow-up time, days, M ($Q_1$, $Q_3$) | 30.00 (18.27, 30.00) | 30.00 (30.00, 30.00) | 9.29 (5.19, 15.79) | Z = -84.773 | <0.001 |
| RDW T0 level, n (%) | | | | $\chi^2$ = 145.970 | <0.001 |
| <14.9% | 3688 (49.19) | 2819 (53.77) | 869 (38.55) | | |
| ≥14.9% | 3809 (50.81) | 2424 (46.23) | 1385 (61.45) | | |
| RDW T1 level, n (%) | | | | $\chi^2$ = 155.500 | <0.001 |
| <15.1% | 3600 (48.02) | 2765 (52.74) | 835 (37.05) | | |
| ≥15.1% | 3897 (51.98) | 2478 (47.26) | 1419 (62.95) | | |
| RDW changes, n (%) | | | | $\chi^2$ = 13.412 | 0.001 |
| <0% | 2187 (29.17) | 1570 (29.94) | 617 (27.37) | | |
| 0%-0.29% | 2518 (33.59) | 1790 (34.14) | 728 (32.30) | | |
| ≥0.3% | 2792 (37.24) | 1883 (35.91) | 909 (40.33) | | |

t: t test, $\chi^2$: chi-square test, Z: rank sum test

ARF: acute respiratory failure, SD: standard deviation, M: median, $Q_1$: 1st quartile, $Q_3$: 3rd quartile, MICU: medical intensive care unit, SICU: surgical intensive care unit, RRT: renal replacement therapy, AKI: acute kidney injury, SBP: systolic blood pressure, DBP: diastolic blood pressure, MAP: mean arterial pressure, RR: respiratory rate, HB: hemoglobin, BUN: blood urea nitrogen, WBC: white blood cell, Cr: creatinine, INR: international normalized ratio, Na: sodium, K: potassium, PT: prothrombin time, PTT: partial thromboplastin time, SOFA: sequential organ failure assessment, CCI: Charlson comorbidity index, RDW: red cell distribution width, LOS: length of stay, RDW T0: RDW value within the 24 hours after ICU admission, RDW T1: RDW value at 24 to 48 hours after ICU admission, RDW changes: RDW T1 minus RDW T0.

predictive performance of SOFA, and SOFA combined with RDW changes on 30-day mortality respectively. In patients with RDW T0 level ≥14.9%, the predictive value of SOFA combined RDW changes on 30-day mortality was a little better than that of SOFA only, with AUCs

**Table 2. Covariates related to the 30-day mortality in ARF.**

| Variables (n = 7497) | Univariable Cox regression model | | Bidirectional stepwise regression model | |
|---|---|---|---|---|
| | HR (95% CI) | *P* | HR (95% CI) | *P* |
| Age | 1.02 (1.02–1.03) | <0.001 | 1.02 (1.01–1.02) | <0.001 |
| Gender | | | | |
| Female | Ref | | | |
| Male | 1.02 (0.94–1.11) | 0.581 | | |
| Ethnicity | | | | |
| Black | Ref | | Ref | |
| White | 1.25 (1.07–1.45) | 0.005 | 1.52 (1.36–1.70) | <0.001 |
| Others | 1.24 (1.03–1.50) | 0.024 | 0.86 (0.74–1.01) | 0.067 |
| Unknown | 1.63 (1.37–1.94) | <0.001 | 1.04 (0.91–1.19) | 0.564 |
| Type of ICU at first hospitalization | | | | |
| MICU | Ref | | Ref | |
| SICU | 0.92 (0.84–1.01) | 0.073 | 1.09 (0.99–1.19) | 0.091 |
| Others | 0.87 (0.77–0.99) | 0.030 | 0.76 (0.67–0.87) | <0.001 |
| Ventilation time | 0.99 (0.99–0.99) | <0.001 | 0.99 (0.99–0.99) | <0.001 |
| Vasopressor use | | | | |
| No | Ref | | Ref | |
| Yes | 1.73 (1.59–1.89) | <0.001 | 1.59 (1.45–1.75) | <0.001 |
| RRT | | | | |
| No | Ref | | Ref | |
| Yes | 1.53 (1.38–1.70) | <0.001 | 1.61 (1.41–1.83) | <0.001 |
| Antiplatelet use | | | | |
| No | Ref | | | |
| Ye | 1.28 (0.53–3.09) | 0.576 | | |
| Antibiotics use | | | | |
| No | Ref | | Ref | |
| Yes | 1.39 (1.20–1.61) | <0.001 | 1.23 (1.05–1.44) | 0.009 |
| Blood transfusions | | | | |
| No | Ref | | | |
| Yes | 1.31 (1.21–1.42) | <0.001 | | |
| Atrial fibrillation | | | | |
| No | Ref | | | |
| Yes | 1.41 (1.29–1.54) | <0.001 | | |
| Pneumonia | | | | |
| No | Ref | | | |
| Yes | 1.04 (0.95–1.13) | 0.374 | | |
| AKI | | | | |
| No | Ref | | Ref | |
| Yes | 2.12 (1.82–2.47) | <0.001 | 1.80 (1.53–2.10) | <0.001 |
| Sepsis | | | | |
| No | Ref | | | |
| Yes | 1.03 (0.94–1.11) | 0.556 | | |
| $SpO_2$ | 1.00 (0.99–1.01) | 0.556 | | |
| MAP | 0.99 (0.99–0.99) | <0.001 | | |
| Temperature | 0.75 (0.71–0.79) | <0.001 | 0.82 (0.78–0.87) | <0.001 |
| Heart rate | 1.00 (1.00–1.00) | 0.099 | | |
| RR | 1.02 (1.01–1.02) | <0.001 | 1.02 (1.01–1.02) | <0.001 |

*(Continued)*

**Table 2.** (Continued)

| Variables (n = 7497) | Univariable Cox regression model | | Bidirectional stepwise regression model | |
|---|---|---|---|---|
| | HR (95% CI) | *P* | HR (95% CI) | *P* |
| Weight | 0.99 (0.99–0.99) | <0.001 | 0.99 (0.99–0.99) | <0.001 |
| FIO$_2$ | 1.01 (1.01–1.01) | 0.003 | 1.01 (1.01–1.01) | 0.017 |
| HB | 0.95 (0.93–0.96) | <0.001 | | |
| BUN | 1.01 (1.01–1.01) | <0.001 | 1.01 (1.01–1.01) | <0.001 |
| Platelet | 0.99 (0.99–0.99) | 0.012 | | |
| WBC | 1.01 (1.01–1.01) | <0.001 | | |
| Hematocrit | 0.99 (0.98–0.99) | <0.001 | | |
| Lactate | 1.10 (1.09–1.12) | <0.001 | 1.05 (1.04–1.07) | <0.001 |
| Glucose | 1.00 (1.00–1.00) | 0.413 | | |
| Cr | 1.03 (1.01–1.05) | 0.002 | 0.84 (0.80–0.88) | <0.001 |
| Bicarbonate | 0.98 (0.97–0.98) | <0.001 | | |
| INR | 1.13 (1.10–1.16) | <0.001 | 1.13 (1.10–1.19) | <0.001 |
| Na | 1.00 (0.99–1.00) | 0.431 | | |
| K | 1.05 (1.01–1.11) | 0.032 | | |
| Chloride | 0.99 (0.99–0.99) | 0.016 | 0.99 (0.99–0.99) | 0.009 |
| PT | 1.01 (1.01–1.02) | <0.001 | 1.01 (1.01–1.01) | 0.002 |
| PTT | 1.01 (1.01–1.01) | <0.001 | 1.01 (1.01–1.01) | 0.008 |
| CCI | 1.15 (1.13–1.16) | <0.001 | 1.11 (1.10–1.13) | <0.001 |

ARF: acute respiratory failure, HR: hazard ratio, CI: confidence interval, Ref: reference, MICU: medical intensive care unit, SICU: surgical intensive care unit, RRT: renal replacement therapy, AKI: acute kidney injury, MAP: mean arterial pressure, RR: respiratory rate, HB: hemoglobin, BUN: blood urea nitrogen, WBC: white blood cell, Cr: creatinine, INR: international normalized ratio, Na: sodium, K: potassium, PT: prothrombin time, PTT: partial thromboplastin time, CCI: Charlson comorbidity index

**Table 3. Association between RDW and RDW changes and 30-day mortality in ARF.**

| Variables (n = 7497) | Univariable model | | Multivariable model | |
|---|---|---|---|---|
| | HR (95% CI) | *P* | HR (95% CI) | *P* |
| RDW T0 | 1.12 (1.11–1.14) | <0.001 | 1.07 (1.05–1.09) | <0.001 |
| RDW T0 level | | | | |
| <14.9% | Ref | | Ref | |
| ≥14.9% | 1.68 (1.54–1.83) | <0.001 | 1.25 (1.15–1.37) | <0.001 |
| RDW changes | 1.13 (1.08–1.19) | <0.001 | 1.08 (1.02–1.13) | 0.003 |
| RDW changes level | | | | |
| <0% | Ref | | Ref | |
| 0%-0.29% | 1.03 (0.93–1.15) | 0.574 | 1.06 (0.95–1.18) | 0.273 |
| ≥0.3% | 1.20 (1.09–1.33) | <0.001 | 1.12 (1.01–1.24) | 0.034 |

RDW: red cell distribution width, ARF: acute respiratory failure, HR: hazard ratio, CI: confidence interval, Ref: reference, RDW T0: RDW value within the 24 hours after ICU admission, RDW T1: RDW value at 24 to 48 hours after ICU admission, RDW changes: RDW T1 minus RDW T0

Multivariable model: adjusted for age, ethnicity, type of ICU at first hospitalization, ventilation time, vasopressor use, RRT, antibiotics use, AKI, temperature, RR, weight, FIO2, BUN, lactate, Cr, INR, chloride, PTT, and CCI.

**Table 4. Association between RDW changes and 30-day mortality in different RDW T0 levels.**

| RDW changes (n = 7497) | RDW T0 <14.9% | | RDW T0 ≥14.9% | |
|---|---|---|---|---|
| | HR (95% CI) | *P* | HR (95% CI) | *P* |
| <0% | Ref | | Ref | |
| 0%-0.29% | 1.12 (0.93–1.35) | 0.244 | 1.09 (0.95–1.25) | 0.222 |
| ≥0.3% | 1.10 (0.91–1.34) | 0.324 | 1.19 (1.05–1.35) | 0.006 |

RDW: red cell distribution width, HR: hazard ratio, CI: confidence interval, Ref: reference, RDW T0: RDW value within the 24 hours after ICU admission, RDW T1: RDW value at 24 to 48 hours after ICU admission, RDW changes: RDW T1 minus RDW T0

Adjusted for age, ethnicity, type of ICU at first hospitalization, ventilation time, vasopressor use, RRT, antibiotics use, AKI, temperature, RR, weight, FIO2, BUN, lactate, Cr, INR, chloride, PTT, and CCI.

of 0.624 vs. 0.620. The sensitivity, negative predictive value, and positive predictive value of SOFA combined RDW changes were respectively 0.563, 0.718, and 0.445.

## Discussion

This study explored the relationship of baseline RDW and its changes and 30-day mortality in patients with ARF. Our results showed that high RDW T0 and RDW changes were both associated with an increased risk of 30-day mortality. These relationships were also found in patients with RDW T0 level ≥14.9%. Furthermore, it seemed that the predictive performance of RDW changes combined with SOFA score on 30-day mortality was a little better than that of SOFA only.

For all we know, no studies have yet focused on the association between RDW and its change during hospitalization with short-term mortality in patients with ARF. Zhang et al. [5] explored the association between RDW and long-term mortality in patients with ARF and found that during the 3 years follow-up, a high RDW on admission was related to an increased risk of long-term mortality. Their results indicated that RDW may be a potential indicator for prognosis and disease progression, which is also easy to get [5]. Wang et al. [15] showed that the RDW level ≥14.5% was an independent predictor for 90-day mortality in patients with acute respiratory distress syndrome (ARDS) comparing to low RDW level. Another propensity scores matched cohort study by Yu et al. [16] also indicated that higher RDW was related to higher 30-day mortality rate in patients with ARDS. Xanthopoulos et al. [17] considered although high RDW values at admission and discharge have been related to adverse prognosis in patients with heart failure, the prognostic role of in-hospital RDW changes remains

**Table 5. Predictive performance of RDW changes on 30-day mortality in different RDW T0 levels.**

| Subgroups (n = 7497) | Variables | AUC (95% CI) | *P* |
|---|---|---|---|
| RDW T0 <14.9% | SOFA | 0.581 (0.559–0.603) | |
| | SOFA + RDW changes | 0.582 (0.560–0.604) | 0.443 |
| RDW T0 ≥14.9% | SOFA | 0.620 (0.601–0.638) | |
| | SOFA + RDW changes | 0.624 (0.605–0.642) | 0.045 |

RDW: red cell distribution width, AUC: areas under curve, CI: confidence interval, Ref: reference, SOFA: sequential organ failure assessment, RDW T0: RDW value within the 24 hours after ICU admission, RDW T1: RDW value at 24 to 48 hours after ICU admission, RDW changes: RDW T1 minus RDW T0

Adjusted for age, ethnicity, type of ICU at first hospitalization, ventilation time, vasopressor use, RRT, antibiotics use, AKI, temperature, RR, weight, FIO2, BUN, lactate, Cr, INR, chloride, PTT, and CCI.

**Table 6. Predictive performance of RDW changes on 30-day mortality in different RDW T0 levels.**

| Subgroups | Variables | Cut-off | AUC (95% CI) | Sensitivity (95% CI) | Specificity (95% CI) | NPV (95% CI) | PPV (95% CI) | Accuracy (95% CI) |
|---|---|---|---|---|---|---|---|---|
| RDW T0 <14.9% | SOFA | 0.6067 | 0.581 (0.559–0.603) | 0.349 (0.317–0.380) | 0.759 (0.743–0.775) | 0.791 (0.775–0.806) | 0.308 (0.279–0.337) | 0.662 (0.647–0.677) |
| | SOFA + RDW changes | 0.4669 | 0.582 (0.560–0.604) | 0.554 (0.520–0.587) | 0.564 (0.546–0.583) | 0.804 (0.786–0.821) | 0.281 (0.260–0.303) | 0.562 (0.546–0.578) |
| RDW T0 ≥14.9% | SOFA | 0.6606 | 0.620 (0.601–0.638) | 0.690 (0.666–0.715) | 0.469 (0.449–0.489) | 0.726 (0.704–0.748) | 0.426 (0.406–0.447) | 0.549 (0.534–0.565) |
| | SOFA + RDW changes | 0.7054 | 0.624 (0.605–0.642) | 0.614 (0.588–0.639) | 0.563 (0.543–0.582) | 0.718 (0.698–0.739) | 0.445 (0.423–0.467) | 0.581 (0.566–0.597) |

RDW: red cell distribution width, AUC: areas under curve, CI: confidence interval, NPV: negative predictive value, PPV: positive predictive value, SOFA: sequential organ failure assessment, RDW T0: RDW value within the 24 h after ICU admission, RDW T1: RDW value at 24 to 48 h after ICU admission, RDW changes: RDW T1 minus RDW T0

Adjusted for age, ethnicity, type of ICU at first hospitalization, ventilation time, vasopressor use, RRT, antibiotics use, AKI, temperature, RR, weight, FIO2, BUN, lactate, Cr, INR, chloride, PTT, and CCI.

debatable. Xiao et al. [18] showed that the measurement of RDW changes has potential to predict the major adverse cardiovascular events in patients with unstable angina underwent percutaneous coronary intervention, and the dynamic changes in RDW were related to the outcome of cardiovascular disease. In this retrospective cohort study in patients with ARF, we found high RDW level at ICU admission and its changes during hospitalization were both associated with the increased risk of 30-day mortality.

RDW reflects the size variation of circulating red blood cells (RBC). Physiologic process that influences the morphology of RBC and causes the early release of young cells into

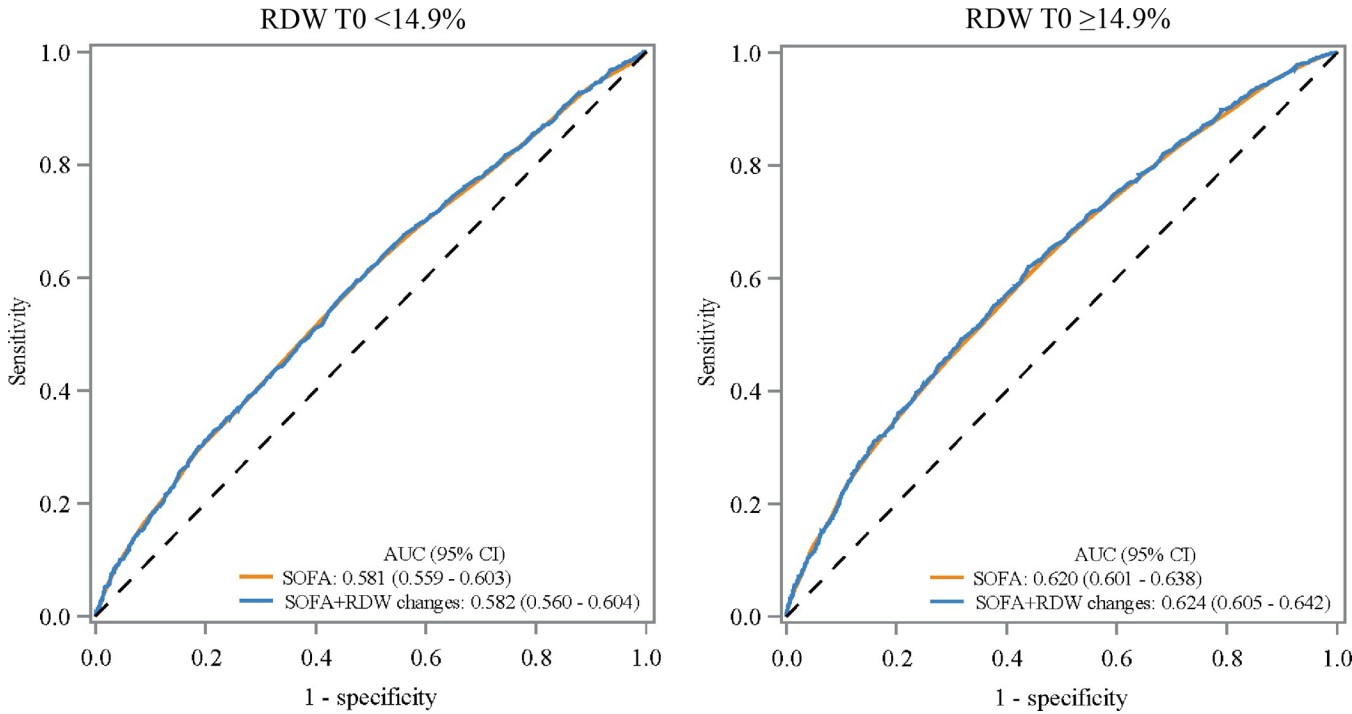

**Fig 2. The ROCs of RDW changes and RDW changes combined with SOFA in subgroups of RDW T0 <14.9% and RDW T0 ≥14.9%.**

circulation can result in the increase of RDW. ARF can lead to severe hypoxemia which induces erythrogenin release, that increases the volume of RBCs leading to an increase in RDW [19, 20]. Abnormal RDW elevation is closely related to proinflammatory factors and consequently associated with the outcomes of several inflammatory diseases [21, 22]. In patients with acute lung injury, oxidative stress can promote the release of young RBCs into the circulation and lead to an increase in RDW [23]. When the numeral value of RDW rised, it means the volume otherness of RBCs increases; when the numeral value of RDW is smaller, on the other side, the volume otherness of RBCs is lower. Therefore, the smaller as the RDW change is, the more stable the RBC volume is, indicating that the RBC response is neutral in disease development [18]. We hypothesized that hypoxemia induced by ARF can promote the synthesis of erythrogenin and further affected RBCs' formation and morphology. Additionally, inflammation and oxidative stress leads to the increase of RDW through promoting a large number of reticulocytes to release into the peripheral circulation due to impaired erythrocyte maturation. Unstable RDW volume (increased RDW) during hospitalization may reflect the rapid progress of inflammation, oxidative stress and liver and kidney function impairment leading to poor ARF prognosis.

Subgroup analyses in patients with different baseline RDW levels also found that RDW changes ≥0.3% was significantly related to an increased risk of 30-day mortality in patients whose RDW T0 level ≥14.9%. RDW at admission is the independent associated factor in many acute diseases. Peng et al. [24] found that after the adjustment for potential confounding factors, RDW at admission remained the independent associated factor with post-stroke fatigue in the acute phase. Cai et al. [25] revealed that RDW ≥14.45% at admission was associated with 28-day mortality in patients with ARDS using cox regression analysis, while Kaplan-Meier analysis showed patients with RDW ≥14.45% had a significantly lower survival rate than those with RDW <14.45%. In our study population, 92.10% ARF patients who died in the hospital had AKI and a higher WBC count compared with the survivors. However, we did not compare inflammation and oxidative stress-related indicators between the survival group and mortality group, and cannot speculate whether elevated baseline RDW levels affect the risk of in-hospital mortality through them. Furthermore, we compared the predictive performance between SOFA and SOFA combined with RDW changes on the 30-day mortality in ARF. The results indicating that taking RDW changes into consideration when using SOFA to predict the ARF prognosis may improve the performance of SOFA score. The SOFA score provides a simple method of assessing and monitoring organ dysfunction in critically ill patients and rapidly became one of the most widely used scoring systems in adult intensive care [26, 27]. In ARF patients, their SOFA scores were respectively 7.00 and 9.00 between survival and 30-day mortality groups, with significantly $FIO_2$ and Cr increasing, and platelet decreasing, indicating respiratory, coagulation, and renal dysfunctions. The results suggested that in clinical practice, RDW change values after admission to the ICU in patients with ARF should be closely monitored and may complement the predictive performance of the SOFA score to some extent and are very easy to obtain. Further studies are needed to explore the predictive value of RDW changes on the prognosis of ARF.

This study was first explored the association between RDW changes and short-term prognosis of ARF, indicating the clinical significance of RDW changes. The baseline RDW and its changes were easy to get that may provide some reference for the early warning of the risk of mortality in ARF patients. There were also some limitations in our research. This was a retrospective study that selection bias was inevitable. The MIMIC-VI database is short of other medication information related to the prognosis of ARF that we could not adjusted in the multivariable model.

## Conclusion

RDW at admission and its changes during the hospitalization were related to the 30-day mortality in ARF patients. Whether RDW changes may help to early identify ARF patients with poor prognosis is needed further explorations.

## Supporting information

**S1 Table. Characteristics of AFR patients before and after the missing data interpolation.**
(DOCX)

**S1 Checklist. STROBE statement—checklist of items that should be included in reports of observational studies.**
(DOCX)

## Author Contributions

**Conceptualization:** Yuyi Shi, Kaihui Sheng.

**Data curation:** Liuxian Shi, Fei Chen, Zhipeng Jiang.

**Formal analysis:** Liuxian Shi, Fei Chen, Zhipeng Jiang.

**Investigation:** Liuxian Shi, Fei Chen, Zhipeng Jiang.

**Methodology:** Liuxian Shi, Fei Chen, Zhipeng Jiang.

**Writing – original draft:** Yuyi Shi.

**Writing – review & editing:** Yuyi Shi, Kaihui Sheng.

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
