## [Decision Letter · Decision Letter 0]

12 Jul 2023

PONE-D-23-16710Association of red cell distribution width and its changes with the 30-day mortality in patients with acute respiratory failure: an analysis of MIMIC-IV databasePLOS ONE

Dear Dr. Sheng,

Thank you for submitting your manuscript to PLOS ONE. After careful consideration, we feel that it has merit but does not fully meet PLOS ONE’s publication criteria as it currently stands. Therefore, we invite you to submit a revised version of the manuscript that addresses the points raised during the review process. 

We look forward to receiving your revised manuscript.

Kind regards,

Vipa Thanachartwet, M.D.

Academic Editor

PLOS ONE

Journal Requirements:

**Additional Editor Comments:**

Please carefully response point by point to the questions raised by the reviewers.

Reviewers' comments:

Reviewer's Responses to Questions

**Comments to the Author**

1. Is the manuscript technically sound, and do the data support the conclusions?

Reviewer #1: Yes

Reviewer #2: Partly

2. Has the statistical analysis been performed appropriately and rigorously? 

Reviewer #1: No

Reviewer #2: Yes

3. Have the authors made all data underlying the findings in their manuscript fully available?

Reviewer #1: Yes

Reviewer #2: Yes

4. Is the manuscript presented in an intelligible fashion and written in standard English?

Reviewer #1: No

Reviewer #2: Yes

5. Review Comments to the Author

Reviewer #1: It's my great honor to have the opportunity to review the manuscript by ：Yuyi Shi et al. Through retrospective study with MIMIC-IV database, they found that high baseline RDW level and its changes during hospitalization was related to the increased risk of 30-day mortality in ARF and RDW changes may be a potential predictor for AKI short-term mortality. The manuscript is interesting and is somehow valuable for the effective management ARF. But there are some concerns need to be explained.

1. RDW T0 was divided into ＜14.9% and ≥14.9%. What is the basis for the grouping.

2. I noticed there is a conceptual error in the sentence “ the predictive value of SOFA combined RDW changes on 30-day mortality was superior to that of SOFA only with AUCs of 1.624 vs. 0.620”, in which on of the AUC arrived at 1.642 (more than 1). How to explain?

3. There are some mistakes in manuscript, such as “AKI” line 46 in page 3 should be “ FRF”; “ ARF one the common diseases -----” in line 50 of page 4 should be “ ARF is one of the ------” .

4. The English needs to be improved.

Reviewer #2: PLOS ONE

July 12, 2023

Manuscript ID: PONE-D-23-16710

Manuscript title: Association of red cell distribution width and its changes with the 30-day mortality in patients with acute respiratory failure: an analysis of MIMIC-IV database

Dear Prof. Vipa Thanachartwet,

This study was conducted as a retrospective cohort study by extract from the Medical Information Mart for Intensive Care IV (MIMIC-IV) database with the aim to explore the association between RDW and its changes and the 30-day mortality in ARF patients. This study concluded that high baseline RDW level and its changes during hospitalization was relate to the increased risk of 30-day mortality in ARF.

However, there are some comments for this manuscript to address as follows:

1. For the ethical concern in the study, this study was conducted as a retrospective cohort study and data were extracted from the Medical Information Mart for Intensive Care IV (MIMIC-IV) Database. The MIMIC was approved by the Institutional Review Boards (IRBs) of BIDMC and the MIT. No ethical approval of our IRB was required since this survey data were publicly available. The question is how to protect privacy information of included patients. This study is an observational study, therefore the authors should state that this study was performed in accordance with the STrengthening the Reporting of OBservational studies in Epidemiology (STROBE) reporting guideline.

2. Study participants, inclusion criteria and exclusion criteria should be clearly defined in Methods.

3. For variable collection, there are a lot of parameters included in the analysis. Some parameters such as SBP, DBP and MAP were related as well as BUN and Cr or etc. The related parameters may cause multicollinearity in the Cox regression analysis.

4. In outcomes and follow-up, the study outcome was 30-day mortality. The MIMIC-III followed up by information in the electronic medical charts and hospital department records, or making contact with the patients, their family members, their attending health care workers, or family physicians on the phone. So, this study was extracted from MIMIC-IV or MIMIC-III Database. How to protect confidentiality of the study participants as “making contact with the patients, their family members, their attending health care workers, or family physicians on the phone”?

5. In statistical analysis,

a. On page 7, “chi-square test was used for cooperation”, what is the meaning of cooperation?

b. On page 8, “Univariable Cox regression and bidirectional stepwise regression analyses were used to screen the covariates related to the 30-day mortality. Univariable and multivariable Cox regression models were established to explore the relationship of RDW T0, RDW T1, and RDW changes and the 30-day mortality in ARF patients with hazard ratios (HRs) and 95% confidence intervals (CIs).” This is rather confusing which method of multivariate analysis was performed in this study.

c. On page 8, “the RDW T0 levels were divided into RDW T0 <14.9% and RDW T0 ≥14.9%.” How and why this cutoff value was selected?

d. On page 8, “We also drew receiver operator characteristic curves (ROCs) with areas under curve (AUCs) to assess the predictive performance of RDW changes combined with SOFA on the 30-day mortality of ARF.” Why RDW changes were combined with SOFA in the analysis?

6. In the results,

a. On page 9, “4,188 (55.86%) of them were females” but “4188 (55.86)” were male in the Table 1.

b. Please revised this sentence “The average RDW T0 between survival and 30-day mortality groups were 15.22% vs. 16.20%, and that of RDW T1 were 15.41% vs. 16.46%.” on page 9 as this may make misinterpreted to the readers.

c. On page 12, what is the meaning of “first care unit”?

d. On page 12, what is the meaning of “significantly linked” and “linked” in the first and second paragraph?

e. On page 16, please check “AUCs of 1.624”.

7. In Table 1,

a. Is it possible to collect all variables of 7,497 patients without any missing data?

b. It is wonder why the “RDW T0 and T1” are expressed as %, Mean ± SD, but the “RDW changes” are expressed as %, M (Q1, Q3).

c. It is wonder why the “RDW T0 (<14.9% vs. ≥14.9%) and RDW T1 (<15.1% vs. ≥15.1%)” have different cutoff values.

d. Please define “LOS” in the footnote of Table 1.

e. The characteristics and baseline data of ARF patients were compared between survival and 30-day mortality group to determine the associated factors for 30-day mortality. It is wonder why all variables were included in the univariate Cox regression model in Table 2.

8. In Table 2,

a. Why MICU was choose as reference in first care unit?

b. Why this “Charlson comorbidity index” was used?

9. Table 4 is unclear.

10. In Table 5, how were “SOFA + RDW changes” combined in the analysis?

11. In Table 6,

a. How to determine the cutoff values?

b. “The predictive value of SOFA combined RDW changes on 30-day mortality was superior to that of SOFA only with AUCs of 1.624 vs. 0.620.” on page 16, but the accuracy of SOFA + RDW changes was only 0.581 (0.566-0.597) as shown in Table 6. Please concern using “superior” in this sentence.

12. Table S1 is not mention in the result.

These are all issues raised for this manuscript and major revision is needed.

6. PLOS authors have the option to publish the peer review history of their article (what does this mean?). If published, this will include your full peer review and any attached files.

Reviewer #1: **Yes: **Feng SHEN

Reviewer #2: **Yes: **Assoc. Prof. Varunee Desakorn

---

## [Author Response · Author response to Decision Letter 0]

31 Aug 2023

Responses to Reviewers

Dear Editor and reviewers,

We highly appreciate and thank the Editors and the reviewers for your time and valuable comments on this manuscript. We have carefully studied the comments and suggestions and revised our paper accordingly. The following are our point-by-point responses to the comments. We hope that the revisions are acceptable and that our responses adequately address the comments. Thank you for your consideration.

Responses to Reviewer #1’s comments

It's my great honor to have the opportunity to review the manuscript by ：Yuyi Shi et al. Through retrospective study with MIMIC-IV database, they found that high baseline RDW level and its changes during hospitalization was related to the increased risk of 30-day mortality in ARF and RDW changes may be a potential predictor for AKI short-term mortality. The manuscript is interesting and is somehow valuable for the effective management ARF. But there are some concerns need to be explained.

1. RDW T0 was divided into <14.9% and ≥14.9%. What is the basis for the grouping.

Response: Thank you for your comment. The RDW T0 was divided into <14.9% level and ≥14.9% level according to the median. Similarly, a previous study based on the MIMIC database that studied RDW was also divided it into low level and high level according to the median. We have added the description for the basis of the grouping in the manuscript.

Mo M, Huang Z, Huo D, Pan L, Xia N, Liao Y, Yang Z. Influence of Red Blood Cell Distribution Width on All-Cause Death in Critical Diabetic Patients with Acute Kidney Injury. Diabetes Metab Syndr Obes. 2022 Aug 2; 15:2301-2309. doi: 10.2147/DMSO.S377650.

2. I noticed there is a conceptual error in the sentence “the predictive value of SOFA combined RDW changes on 30-day mortality was superior to that of SOFA only with AUCs of 1.624 vs. 0.620”, in which on of the AUC arrived at 1.642 (more than 1). How to explain?

Response: Thank you for your comment. We are so sorry about this clerical error. The actual AUC value was 0.624, and we have revised it.

3. There are some mistakes in manuscript, such as “AKI” line 46 in page 3 should be “FRF”; “ARF one the common diseases -----” in line 50 of page 4 should be “ARF is one of the ------”.

Response: Thank you for your comment. We have changed “AKI” into “ARF”, “ARF one the common diseases -----” into “ARF is one of the ------”

4. The English needs to be improved.

Response: Thank you for your comment. We have carefully checked the manuscript, and revised the mistakes to improve our English writing. Also, we invited a native English speaker to help us revised this article.

Responses to Reviewer #2’s comments

Dear Prof. Vipa Thanachartwet,

This study was conducted as a retrospective cohort study by extract from the Medical Information Mart for Intensive Care IV (MIMIC-IV) database with the aim to explore the association between RDW and its changes and the 30-day mortality in ARF patients. This study concluded that high baseline RDW level and its changes during hospitalization was relate to the increased risk of 30-day mortality in ARF.

However, there are some comments for this manuscript to address as follows:

1. For the ethical concern in the study, this study was conducted as a retrospective cohort study and data were extracted from the Medical Information Mart for Intensive Care IV (MIMIC-IV) Database. The MIMIC was approved by the Institutional Review Boards (IRBs) of BIDMC and the MIT. No ethical approval of our IRB was required since this survey data were publicly available. The question is how to protect privacy information of included patients. This study is an observational study; therefore, the authors should state that this study was performed in accordance with the STrengthening the Reporting of OBservational studies in Epidemiology (STROBE) reporting guideline.

Response: Thank you for your comment. The MIMIC-IV database was publicly available and was approved by the IRBs of BIDMC and the MIT. How the MIMIC-IV to protect privacy information of included patients may refer to the relevant website description: https://www.physionet.org/content/mimiciv/2.2/. We also revised the ethical statement to clarify this study was performed in accordance with the STROBE reporting guideline.

2. Study participants, inclusion criteria and exclusion criteria should be clearly defined in Methods.

Response: Thank you for your comment. We have revised the Methods section to clearly define the study participants, inclusion criteria and exclusion criteria.

3. For variable collection, there are a lot of parameters included in the analysis. Some parameters such as SBP, DBP and MAP were related as well as BUN and Cr or etc. The related parameters may cause multicollinearity in the Cox regression analysis.

Response: Thank you and we are agree with your comment. We have performed the multicollinearity test on variables including age, ventilation time, temperature, respiratory rate, weight, FIO2, BUN, lactate, creatinine, chloride, PT, PTT, and CCI. The results were showed in the following table, and all the variance inflation factors (VIFs) all approached 1.

Variables VIF

age 1.000293026

ventilation time 1.00000669

temperature 1.040151796

respiratory rate 1.000236109

weight 1.000050414

FIO2 1.00000495

BUN 1.000059585

lactate 1.002710493

creatinine 1.032349135

chloride 1.000057775

PT 1.000030777

PTT 1.000004819

CCI 1.011766789

4. In outcomes and follow-up, the study outcome was 30-day mortality. The MIMIC-III followed up by information in the electronic medical charts and hospital department records, or making contact with the patients, their family members, their attending health care workers, or family physicians on the phone. So, this study was extracted from MIMIC-IV or MIMIC-III Database. How to protect confidentiality of the study participants as “making contact with the patients, their family members, their attending health care workers, or family physicians on the phone”?

Response: Thank you and we are agree with your comment. The MIMIC-IV aims to carry on the success of MIMIC-III, which adopted a permissive access scheme which allowed for broad reuse of the data. The protection for the confidentiality of the study participants run through the creation of MIMIC-IV carried out steps. Patient identifiers as stipulated by HIPAA were removed. Patient identifiers were replaced using a random cipher, resulting in deidentified integer identifiers for patients, hospitalizations, and ICU stays. Structured data were filtered using look up tables and allow lists.

1. Johnson, A. E., Pollard, T. J., Shen, L., Lehman, L.H., Feng, M., Ghassemi, M., ... & Mark, R. G. (2016). MIMIC-III, a freely accessible critical care database. Scientific data, 3(1), 1-9.

5. In statistical analysis,

a. On page 7, “chi-square test was used for cooperation”, what is the meaning of cooperation?

Response: Thank you for your comment. We are so sorry for the clerical error, and we have revised the “cooperation” into “comparation.”

b. On page 8, “Univariable Cox regression and bidirectional stepwise regression analyses were used to screen the covariates related to the 30-day mortality. Univariable and multivariable Cox regression models were established to explore the relationship of RDW T0, RDW T1, and RDW changes and the 30-day mortality in ARF patients with hazard ratios (HRs) and 95% confidence intervals (CIs).” This is rather confusing which method of multivariate analysis was performed in this study.

Response: Thank you for your comment. We have revised the description for the methods of statistical analyses.

c. On page 8, “the RDW T0 levels were divided into RDW T0 <14.9% and RDW T0 ≥14.9%.” How and why this cutoff value was selected?

Response: Thank you for your comment. The RDW T0 was divided into <14.9% level and ≥14.9% level according to the median. Similarly, a previous study based on the MIMIC database that studied RDW was also divided it into low level and high level according to the median. We have added the description for the basis of the grouping in the manuscript.

Mo M, Huang Z, Huo D, Pan L, Xia N, Liao Y, Yang Z. Influence of Red Blood Cell Distribution Width on All-Cause Death in Critical Diabetic Patients with Acute Kidney Injury. Diabetes Metab Syndr Obes. 2022 Aug 2; 15:2301-2309. doi: 10.2147/DMSO.S377650.

d. On page 8, “We also drew receiver operator characteristic curves (ROCs) with areas under curve (AUCs) to assess the predictive performance of RDW changes combined with SOFA on the 30-day mortality of ARF.” Why RDW changes were combined with SOFA in the analysis?

Response: Thank you for your comment. The SOFA score is a validated prognostic score ranging from 0–24, with points assigned for evidence of organ failure within 6 different organ systems, with higher scores correlating with a higher likelihood of in-hospital mortality. This study aimed to explore the association between RDW and its changes and the 30-day mortality in ARF. Also, we assessed the predictive performance of RDW changes on 30-day mortality in ARF, using the SOFA as a comparison object. Xu et al. developed and validated a nomogram to predict the mortality risk in elderly patients with ARF, in which they similarly compared the predictive performance of their nomogram with SOFA score.

Ferreira FL, Bota DP, Bross A, Mélot C, Vincent J-L. Serial evaluation of the SOFA score to predict outcome in critically ill patients. JAMA. 2001;286(14):1754–8. doi: 10.1001/jama.286.14.1754

Raith EP, Udy AA, Bailey M, McGloughlin S, MacIsaac C, Bellomo R, et al. Prognostic accuracy of the SOFA score, SIRS criteria, and qSOFA score for in-hospital mortality among adults with suspected infection admitted to the intensive care unit. JAMA. 2017;317(3):290–300. doi: 10.1001/jama.2016.20328

Xu J, Weng J, Yang J, Shi X, Hou R, Zhou X, Zhou Z, Wang Z, Chen C. Development and validation of a nomogram to predict the mortality risk in elderly patients with ARF. PeerJ. 2021 Mar 9;9:e11016. doi: 10.7717/peerj.11016. PMID: 33854838; PMCID: PMC7953875.

6. In the results,

a. On page 9, “4,188 (55.86%) of them were females” but “4188 (55.86)” were male in the Table 1.

Response: Thank you for your comment. We are so sorry for the clerical error, and we have revised the “females” into “males.”

b. Please revised this sentence “The average RDW T0 between survival and 30-day mortality groups were 15.22% vs. 16.20%, and that of RDW T1 were 15.41% vs. 16.46%.” on page 9 as this may make misinterpreted to the readers.

Response: We have revised this sentence.

c. On page 12, what is the meaning of “first care unit”?

Response: Thank you for your comment. The variable “first care unit” mean the type of the ICU when the patients hospitalized in at the first time, including MICU, SICU and others. To make it clear, we have changed the “first care unit” into “type of ICU at first hospitalization”.

d. On page 12, what is the meaning of “significantly linked” and “linked” in the first and second paragraph?

Response: Thank you for your comment. We have revised the “linked to” into “related to.”

e. On page 16, please check “AUCs of 1.624”.

Response: Thank you for your comment. We are so sorry about this clerical error. The actual AUC value was 0.624, and we have revised it.

7. In Table 1,

a. Is it possible to collect all variables of 7,497 patients without any missing data?

Response: Thank you for your comment. The study data were extracted from the MIMIC-IV database, it was not realistic to collect all variables of 7,497 patients without any missing data. However, the missing data which proportion over 20% were deleted, otherwise were interpolated by the random forest method. In addition, we performed the sensitivity analysis of the participants’ characteristics before and after the interpolation, and the results showed that there was no significant difference them.

b. It is wonder why the “RDW T0 and T1” are expressed as %, Mean ± SD, but the “RDW changes” are expressed as %, M (Q1, Q3).

Response: Thank you for your comment. We have mentioned in the Statistical analysis section that normal distributed data were described by Mean ± SD, while skewed distribution data were described by M (Q1, Q3). The RDW T0 and T1 were normal distributed data, and the RDW change was skewed distribution data.

c. It is wonder why the “RDW T0 (<14.9% vs. ≥14.9%) and RDW T1 (<15.1% vs. ≥15.1%)” have different cutoff values.

Response: Thank you for your comment. We divided the RDW T0 and T1 into two levels respectively according to their own median. The median of RDW T0 among participants was 14.9%, while the median of RDW T1 was 15.1%.

d. Please define “LOS” in the footnote of Table 1.

Response: Thank you for your comment. We have added the definition of “LOS” in the footnote.

e. The characteristics and baseline data of ARF patients were compared between survival and 30-day mortality group to determine the associated factors for 30-day mortality. It is wonder why all variables were included in the univariate Cox regression model in Table 2.

Response: Thank you for your comment. Table 2 showed the process of covariates screening. We used univariable Cox regression analysis and bidirectional stepwise regression analysis to screening the covariates related to 30-day mortality in ARF. All the variables were included in the univariable Cox regression analysis, and those with P values <0.05 were considered the potential covariates that were then included into the bidirectional stepwise regression analysis. Finally, variables with P values <0.05 in the bidirectional stepwise regression analysis were the covariates for further adjustment.

8. In Table 2,

a. Why MICU was choose as reference in first care unit?

Response: Thank you for your comment. Table 2 showed the process of covariates screening. When take classified variables into consideration, we picked one of the categories at random as the reference. In other words, it is not obligatory to use the “MICU” as a reference, we can also choose “SICU” or “others” as the reference.

b. Why this “Charlson comorbidity index” was used?

Response: Thank you for your comment. The Charlson comorbidity index (CCI) reflects the severity of illness, which may influence the 30-day mortality in patients with ARF. We included it into the covariates screening, and found it was significantly associated with the 30-day mortality. In turn, when we explored the association between RDW and its changes and the 30-day mortality in patients with ARF, we adjusted for the CCI.

Tuty Kuswardhani RA, Henrina J, Pranata R, Anthonius Lim M, Lawrensia S, Suastika K. Charlson comorbidity index and a composite of poor outcomes in COVID-19 patients: A systematic review and meta-analysis. Diabetes Metab Syndr. 2020 Nov-Dec;14(6):2103-2109. doi: 10.1016/j.dsx.2020.10.022.

9. Table 4 is unclear.

Response: Thank you for your comment. We have revised the Table 4.

10. In Table 5, how were “SOFA + RDW changes” combined in the analysis?

Response: Thank you for your comment. We compared the predictive performance of SOFA combined with RDW changes with that of only SOFA score on 30-day mortality in patients with ARF. The “SOFA + RDW changes” was regarded as a multivariable model which was composed of variables including “SOFA” and “RDW changes”. We aimed to explore whether the predictive value of SOFA score combined with RDW changes on the 30-day mortality in patients with ARF is better than that of single SOFA score, in order to assess the prognostic value and complementary significance of RDW to the existing scoring system in clinical.

11. In Table 6,

a. How to determine the cutoff values?

Response: Thank you for your comment. The cutoff value was calculated using the Youden index, which was a summary measure widely used in the evaluation of the diagnostic accuracy of a medical test.

Habibzadeh F, Habibzadeh P, Yadollahie M. On determining the most appropriate test cut-off value: the case of tests with continuous results. Biochem Med (Zagreb). 2016 Oct 15;26(3):297-307. doi: 10.11613/BM.2016.034.

b. “The predictive value of SOFA combined RDW changes on 30-day mortality was superior to that of SOFA only with AUCs of 1.624 vs. 0.620.” on page 16, but the accuracy of SOFA + RDW changes was only 0.581 (0.566-0.597) as shown in Table 6. Please concern using “superior” in this sentence. 

Response: Thank you for your comment. We have revised the “superior to” into “a little better than.”

12. Table S1 is not mention in the result.

These are all issues raised for this manuscript and major revision is needed.

Response: Thank you for your comment. We have added the Table S1 in the result section.

---

## [Decision Letter · Decision Letter 1]

20 Sep 2023

PONE-D-23-16710R1Association of red cell distribution width and its changes with the 30-day mortality in patients with acute respiratory failure: an analysis of MIMIC-IV databasePLOS ONE

Dear Dr. Sheng Kaihui,

Thank you for submitting your manuscript to PLOS ONE. After careful consideration, we feel that it has merit but does not fully meet PLOS ONE’s publication criteria as it currently stands. Therefore, we invite you to submit a revised version of the manuscript that addresses the points raised during the review process.

We appreciate the interesting and well conducted study. However, there are some minor points raised by the reviewers. Please carefully respond to the reviewer comments and suggestions.

We look forward to receiving your revised manuscript.

Kind regards,

Vipa Thanachartwet, M.D.

Academic Editor

PLOS ONE

Journal Requirements:

Reviewers' comments:

Reviewer's Responses to Questions

**Comments to the Author**

1. If the authors have adequately addressed your comments raised in a previous round of review and you feel that this manuscript is now acceptable for publication, you may indicate that here to bypass the “Comments to the Author” section, enter your conflict of interest statement in the “Confidential to Editor” section, and submit your "Accept" recommendation.

Reviewer #1: All comments have been addressed

Reviewer #2: All comments have been addressed

2. Is the manuscript technically sound, and do the data support the conclusions?

Reviewer #1: Yes

Reviewer #2: Partly

3. Has the statistical analysis been performed appropriately and rigorously? 

Reviewer #1: Yes

Reviewer #2: Yes

4. Have the authors made all data underlying the findings in their manuscript fully available?

Reviewer #1: Yes

Reviewer #2: Yes

5. Is the manuscript presented in an intelligible fashion and written in standard English?

Reviewer #1: Yes

Reviewer #2: Yes

6. Review Comments to the Author

Reviewer #1: In table 3，4, it included “RDW changes ＜0%”. What dose it mean by RDW changes ＜0%? The minimal rate of change for any indicator is zero and should not be less than zero. Please explain.

Reviewer #2: PLOS ONE

September 16, 2023

Manuscript ID: PONE-D-23-16710

Manuscript title: Association of red cell distribution width and its changes with the 30-day mortality in patients with acute respiratory failure: an analysis of MIMIC-IV database

Dear Prof. Vipa Thanachartwet,

The revised manuscript is much improved. However, there are minor comments for this manuscript to address as follows:

1. In outcomes and follow-up (page 7, line 125), in order to protect confidentiality of the study participants, method of deidentification of study participants in the responses to the reviewers should be added in this section.

2. In page 8, line 149, how important of SOFA score should be added from response to the reviewers.

3. Table S1 is disappeared.

4. In the results (page 9, line 169-170), this is only descriptive, but no interpretation of results.

5. Table 1 and Table 3, RDW T0, RDW T1, and RDW changes was classified into categorical variables, but these are not mention in the results. The categorical data of RDW changes was overlapping (<0%, 0-0.3%, ≥0.3%). If the data was 0.3%, how to classify.

6. In the results (page 13, line 199), “High RDW T0” did not mention before, therefore it is difficult to follow.

7. Table 2-6, there was no “n” number of sample for each variable.

8. The description of results for Table 4 (page 15, line 220-222) was unclear.

9. Table 6, How to determine the cutoff values?

10. In discussion (page 21, line 309-310), it is wonder that the study population were patients with ARF, how to compare AKI and high WBC with those without ARF.

These are all issues raised for this manuscript and minor revision is needed.

7. PLOS authors have the option to publish the peer review history of their article (what does this mean?). If published, this will include your full peer review and any attached files.

Reviewer #1: **Yes: **Feng SHEN

Reviewer #2: **Yes: **Assoc Prof. Varunee Desakorn

---

## [Author Response · Author response to Decision Letter 1]

7 Oct 2023

Responses to Reviewers

Dear Editor and reviewers,

We highly appreciate and thank the Editors and the reviewers for your time and valuable comments on this manuscript. We have carefully studied the comments and suggestions and revised our paper accordingly. The following are our point-by-point responses to the comments. We hope that the revisions are acceptable and that our responses adequately address the comments. Thank you for your consideration.

Responses to Reviewer #1’s comments

In table 3, 4, it included “RDW changes <0%”. What does it mean by RDW changes <0%? The minimal rate of change for any indicator is zero and should not be less than zero. Please explain.

Response: Thank you for your comment. In this study, RDW change values were not the absolute value. The RDW changes were calculated by RDW T1 - RDW T0, and when RDW level at T0 higher than that at T1, the value of RDW change was <0%.

Responses to Reviewer #2’s comments

The revised manuscript is much improved. However, there are minor comments for this manuscript to address as follows:

1. In outcomes and follow-up (page 7, line 125), in order to protect confidentiality of the study participants, method of deidentification of study participants in the responses to the reviewers should be added in this section.

Response: Thank you for your valuable comment. We have added the description of deidentification of study participants in this section.

2. In page 8, line 149, how important of SOFA score should be added from response to the reviewers.

Response: Thank you for your valuable comment. We have added the importance of SOFA score from the response to the reviewers.

3. Table S1 is disappeared.

Response: Thank you for your comment. We are very sorry for the mistake, and we have re-uploaded the Table S1.

4. In the results (page 9, line 169-170), this is only descriptive, but no interpretation of results.

Response: Thank you for your valuable comment. We have revised this paragraph into description for the interpretation of results. Table 1 showed the characteristics of participants, and the comparation between groups was base on the crude model, which we think could not indicate the real association between the baseline variables and the study outcome.

5. Table 1 and Table 3, RDW T0, RDW T1, and RDW changes was classified into categorical variables, but these are not mention in the results. The categorical data of RDW changes was overlapping (<0%, 0-0.3%, ≥0.3%). If the data was 0.3%, how to classify.

Response: Thank you for your valuable comment. We have added the description of RDW T0, RDW T1, and RDW changes as categorical variables in the methods section. Also, the intervals of RDW changes have been revised into <0%, 0%-0.29%, and ≥0.3%.

6. In the results (page 13, line 199), “High RDW T0” did not mention before, therefore it is difficult to follow.

Response: Thank you for your valuable comment. We have revised this sentence to make it clear.

7. Table 2-6, there was no “n” number of sample for each variable.

Response: Thank you for your comment. The number of samples for each variable has been added in these tables.

8. The description of results for Table 4 (page 15, line 220-222) was unclear.

Response: Thank you for your valuable comment. We have revised the description of results for Table 4.

9. Table 6, How to determine the cutoff values?

Response: Thank you for your comment. The cutoff value was calculated by the Youden index, which reflects the optimum cutoff of the prediction probability. More details of the ROC curve and Youden index could reference the following article. Also, in our study, we mainly used the AUC, sensitivity, and specificity of ROC curves to assess the predictive performance of RDW changes on 30-day mortality in ARF.

Bantis LE, Tsimikas JV, Chambers GR, Capello M, Hanash S, Feng Z. The length of the receiver operating characteristic curve and the two cutoff Youden index within a robust framework for discovery, evaluation, and cutoff estimation in biomarker studies involving improper receiver operating characteristic curves. Stat Med. 2021 Mar 30;40(7):1767-1789. doi: 10.1002/sim.8869.

10. In discussion (page 21, line 309-310), it is wonder that the study population were patients with ARF, how to compare AKI and high WBC with those without ARF.

These are all issues raised for this manuscript and minor revision is needed.

Response: Thank you for your comment. We are very sorry for this clerical error. The comparation has been between survival group and 30-day mortality group in this study. We have revised this sentence into “In our study population, 92.10% ARF patients who died in the hospital had AKI and a higher WBC count compared with the survivors.”

---

## [Decision Letter · Decision Letter 2]

24 Oct 2023

Association of red cell distribution width and its changes with the 30-day mortality in patients with acute respiratory failure: an analysis of MIMIC-IV database

PONE-D-23-16710R2

Dear Dr. Kaihui Sheng,

We’re pleased to inform you that your manuscript has been judged scientifically suitable for publication and will be formally accepted for publication once it meets all outstanding technical requirements.

Kind regards,

Vipa Thanachartwet, M.D.

Academic Editor

PLOS ONE

Additional Editor Comments (optional):

All comments have been addressed.

Reviewers' comments:

Reviewer's Responses to Questions

**Comments to the Author**

1. If the authors have adequately addressed your comments raised in a previous round of review and you feel that this manuscript is now acceptable for publication, you may indicate that here to bypass the “Comments to the Author” section, enter your conflict of interest statement in the “Confidential to Editor” section, and submit your "Accept" recommendation.

Reviewer #1: All comments have been addressed

Reviewer #2: All comments have been addressed

2. Is the manuscript technically sound, and do the data support the conclusions?

Reviewer #1: Yes

Reviewer #2: Yes

3. Has the statistical analysis been performed appropriately and rigorously? 

Reviewer #1: Yes

Reviewer #2: Yes

4. Have the authors made all data underlying the findings in their manuscript fully available?

Reviewer #1: No

Reviewer #2: Yes

5. Is the manuscript presented in an intelligible fashion and written in standard English?

Reviewer #1: Yes

Reviewer #2: Yes

6. Review Comments to the Author

Reviewer #1: (No Response)

Reviewer #2: (No Response)

7. PLOS authors have the option to publish the peer review history of their article (what does this mean?). If published, this will include your full peer review and any attached files.

Reviewer #1: **Yes: **Feng SHEN

Reviewer #2: **Yes: **Assoc. Prof. Varunee Desakorn

---

## [Editor Report · Acceptance letter]

26 Oct 2023

PONE-D-23-16710R2 

Association of red cell distribution width and its changes with the 30-day mortality in patients with acute respiratory failure: an analysis of MIMIC-IV database 

Dear Dr. Sheng:

I'm pleased to inform you that your manuscript has been deemed suitable for publication in PLOS ONE. Congratulations! Your manuscript is now with our production department. 

Kind regards, 

on behalf of

Professor Vipa Thanachartwet 

Academic Editor

PLOS ONE